# CHARTREASONER: CODE-DRIVEN MODALITY BRIDGING FOR LONG-CHAIN REASONING IN CHART QUESTION ANSWERING

## ABSTRACT

Recently, large language models have shown remarkable reasoning capabilities through long-chain reasoning before responding. However, how to extend this capability to visual reasoning tasks remains an open challenge. Existing multimodal reasoning approaches transfer such visual reasoning task into textual reasoning task via several image-to-text conversions, which often lose critical structural and semantic information embedded in visualizations, especially for tasks like chart question answering that require a large amount of visual details. To bridge this gap, we propose ChartReasoner, a code-driven novel two-stage framework designed to enable precise, interpretable reasoning over charts. We first train a high-fidelity model to convert diverse chart images into structured ECharts codes, preserving both layout and data semantics as lossless as possible. Then, we design a general chart reasoning data synthesis pipeline, which leverages this pretrained transport model to automatically and scalably generate chart reasoning trajectories and utilizes a code validator to filter out low-quality samples. Finally, we train the final multimodal model using a combination of supervised fine-tuning and reinforcement learning on our synthesized chart reasoning dataset and experimental results on four public benchmarks clearly demonstrate the effectiveness of our proposed ChartReasoner. It can preserve the original details of the charts as much as possible and perform comparably with state-of-the-art open-source models while using fewer parameters, approaching the performance of proprietary systems like GPT-4o in out-of-domain settings.

## 1 INTRODUCTION

Chart question answering (ChartQA) aims to enable models to understand and reason over structured visualizations such as bar and line charts. Recent models have improved visual-text alignment (Masry et al., 2023; Liu et al., 2023), while ChartLlama (Han et al., 2023) and ChartSFT (Meng et al., 2024) introduce chain-of-thought (CoT) prompting for multi-step reasoning. However, most existing ChartQA models still lack true reasoning capabilities. The CoT reasoning they produce is often shallow and short, resulting in superficial reasoning without genuine logical depth.

Large language models (LLMs) have achieved remarkable success in text-based long-chain reasoning, producing highly accurate, structured, and multi-step solutions to complex problems. This is exemplified by LLMs such as DeepSeek-R1 (Guo et al., 2025). These models decompose complex problems into logical sequential steps, each building upon previous deductions to reach well-justified conclusions. However, this reasoning capability remains largely confined to the textual domain, creating a significant gap when applied to visual chart interpretation tasks. In particular, ChartQA poses unique challenges that cannot be addressed by simple extensions of text-based reasoning. Existing multimodal approaches typically convert visual inputs into textual representations via image-to-text pipelines. While effective in some contexts, this strategy often fails to preserve the structural and semantic fidelity of the original visualizations. As a result, critical layout, spatial, and quantitative details necessary for accurate reasoning in ChartQA are frequently lost, severely limiting the effectiveness of current models in this domain.

Recent advances in multimodal reasoning extend structured thinking from text to vision by converting images into textual representations to enable CoT reasoning. Methods such as R1-OneVision (Chen et al., 2025) translate visual scenes into formal text, while R1-V (Chen et al., 2025), Curr-ReFT (Deng et al., 2025), LMM-R1 (Peng et al., 2025) and MMEureka (Meng et al., 2025) leverage reinforcement learning to enhance object-centric and long chain reasoning. Despite these innovations, visual content is often treated as auxiliary, serialized into language at the cost of losing fine-grained cues. Local structures, color semantics, and spatial layouts are frequently abstracted or compressed. This lossy transformation undermines tasks that require precise visual grounding, such as ChartQA or scientific diagram analysis.

To address the challenges of chart-based understanding and long-chain reasoning, we propose ChartReasoner, a code-driven, two-stage framework that enhances the reasoning capabilities of multimodal large language models (MLLMs). In the first stage, we train Chart2Code, a high-accuracy model that translates diverse chart images into structured ECharts code, faithfully preserving both visual layout and underlying data semantics. This symbolic representation serves as the foundation for reasoning, bridging the visual–textual modality gap. In the second stage, we construct the Chart-Think dataset by applying Chart2Code to various benchmarks, yielding 140K multi-step reasoning samples. We then train the final ChartReasoner model through supervised fine-tuning (SFT) and reinforcement learning (RL) to improve reasoning accuracy, consistency, and interpretability. This structured pipeline enables precise, scalable, and logically grounded ChartQA.

Contributions are as follows:

- We introduce ChartThink, the first large-scale chart reasoning dataset with over 140K multi-step samples, supporting interpretable and logic-driven analysis across diverse chart types. We also construct Chart2Code, a dataset of 110K synthetic charts paired with accurate ECharts code, bridging the visual–textual modality gap.
- We propose ChartReasoner, a two-stage model that first translates chart images into symbolic ECharts code using Chart2Code, then performs multi-step reasoning on the structured representation. This improves accuracy and generalization across ChartQA tasks.
- ChartReasoner is comparable to state-of-the-art open-source models on ChartQA, ChartBench, EvoChart-QA, and ChartQAPro, while using fewer parameters, and approaches the performance of proprietary systems like GPT-4o in out-of-domain settings.

## 2 RELATED WORK

**ChartQA.** To enhance MLLMs' chart understanding, datasets like ChartQA and others have been proposed (Kahou et al., 2017; Kafle et al., 2018; Methani et al., 2020; Chaudhry et al., 2020; Masry et al., 2022), covering diverse chart types and visual reasoning tasks. However, most focus on single-value or label-based answers and lack support for complex multi-step reasoning. Recent efforts such as ChartX and related work (Xia et al., 2024; Ahmed et al., 2023; Masry et al., 2023; Meng et al., 2024; Huang et al., 2025a) scale up data via synthetic generation, template-based QA, and CoT annotations, though reasoning depth remains limited. On the model side, compact models like ChartReader and others (Cheng et al., 2023; Liu et al., 2023; Baechler et al., 2024; Masry et al., 2023) show strong results on early benchmarks. LLaVA-based models including ChartLlama and its variants (Han et al., 2023; Carbune et al., 2024; Masry et al., 2024; Meng et al., 2024; Zhang et al., 2024) further enhance multimodal alignment. More recently, generalist vision-language models like Phi-3 Vision (Chen et al., 2024a) have also achieved promising performance. Nevertheless, current models still struggle with long-chain reasoning involving multi-cue integration and numerical-logical inference.

**Multimodal Long-Chain Reasoning.** Long-chain reasoning has gained momentum in NLP with the emergence of DeepSeek-R1 (Guo et al., 2025), which emphasizes structured intermediate steps. This paradigm has been extended to vision-language models (VLMs) through works like R1-OneVision (Yang et al., 2025b) and Vision-R1 (Huang et al., 2025b), which convert images into formal textual representations to enable multimodal CoT training. R1-V (Chen et al., 2025) applies Group Relative Policy Optimization (GRPO) (Shao et al., 2024) to object counting, demonstrating that small models can outperform larger ones with effective reinforcement learning. VisualThinker-R1-Zero (Zhou et al., 2025), Curr-ReFT (Deng et al., 2025), LMM-R1 (Peng et al., 2025), and

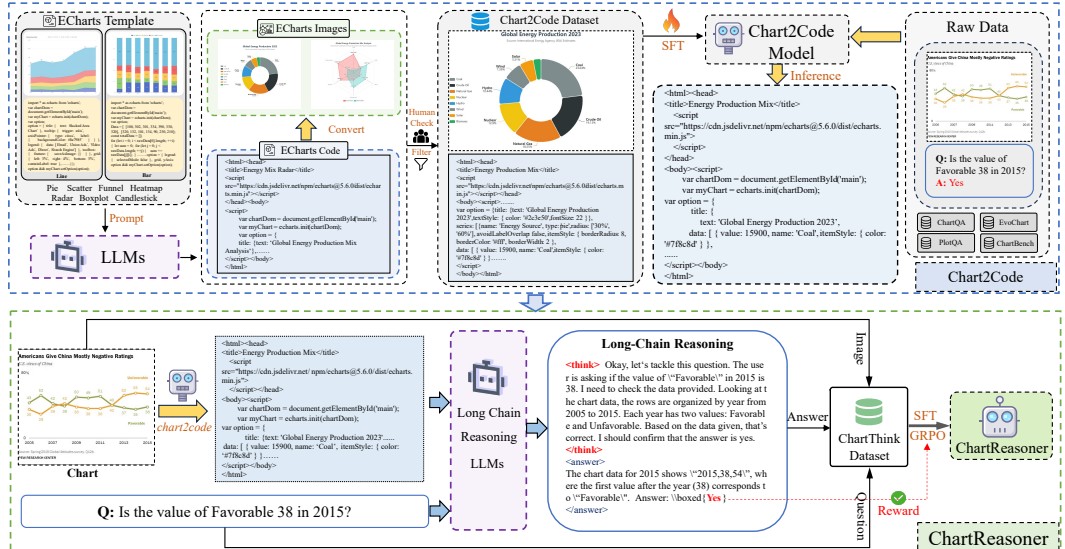

Figure 1: Overview of the data construction pipeline and model training. (1) Chart2Code: We generate a synthetic dataset by rendering chart specifications into paired images and structured code using a prompt-based pipeline. A hybrid filtering strategy is applied to ensure data quality. A vision-language model is then fine-tuned on these image–code pairs. (2) ChartReasoner: Existing ChartQA datasets are converted into structured code using the trained Chart2Code model. Reasoning traces are then generated over code representations to build a symbolic reasoning dataset. The final model is trained in two stages.

MMEureka (Meng et al., 2025) further explore RL-driven reasoning, revealing "visual aha moments" where longer outputs indicate deeper reasoning. Yet most methods still target natural or generic images and remain challenged by structured inputs such as charts.

To address the challenges of chart understanding and long-chain reasoning, we propose ChartReasoner, a code-driven, two-stage framework that enhances the reasoning capabilities of MLLMs. In stage one, Chart2Code translates diverse chart images into structured ECharts code, preserving visual layouts and data semantics. Compared with prior methods such as ChartMimic (Yang et al., 2025a), Plot2Code (Wu et al., 2025), and ChartX (Xia et al., 2024), which focus on layout heuristics or require costly supervision (Li et al., 2025; Zhang et al., 2025), Chart2Code offers greater flexibility and fidelity via executable, symbolic code. While ChartCoder (Zhao et al., 2025) adopts a similar code-centric approach, its reliance on fixed templates limits generalization. In stage two, we generate 140K multi-step reasoning samples by applying Chart2Code to existing ChartQA benchmarks, forming the ChartThink dataset. ChartReasoner is then trained with supervised and reinforcement learning, yielding accurate, consistent, and interpretable reasoning. This structured pipeline bridges the visual–textual gap and supports logically grounded ChartQA.

## 3 OUR METHOD

Understanding charts remains a core challenge for MLLMs due to the gap between visual inputs and symbolic semantics. To bridge this gap, we propose a code-driven, two-stage framework that integrates visual perception with symbolic reasoning through structured chart representations. In the first stage, we introduce Chart2Code, a model that converts chart images into executable ECharts code, preserving both visual layout and semantic structure. To train it, we generate a 110K synthetic dataset via a prompt-based pipeline, rendering chart specifications into image–code pairs and applying a hybrid filtering strategy to ensure quality. The model is fine-tuned with a frozen visual encoder and a trainable decoder for accurate symbolic generation. In the second stage, we use Chart2Code to construct ChartThink, a 140K dataset of multi-step reasoning samples. These are derived by extracting structured code from existing ChartQA benchmarks and prompting a language model to generate chain-of-thought reasoning over the code. This symbolic abstraction enables precise and lossless reasoning over chart semantics. Our final model, ChartReasoner, is trained in two

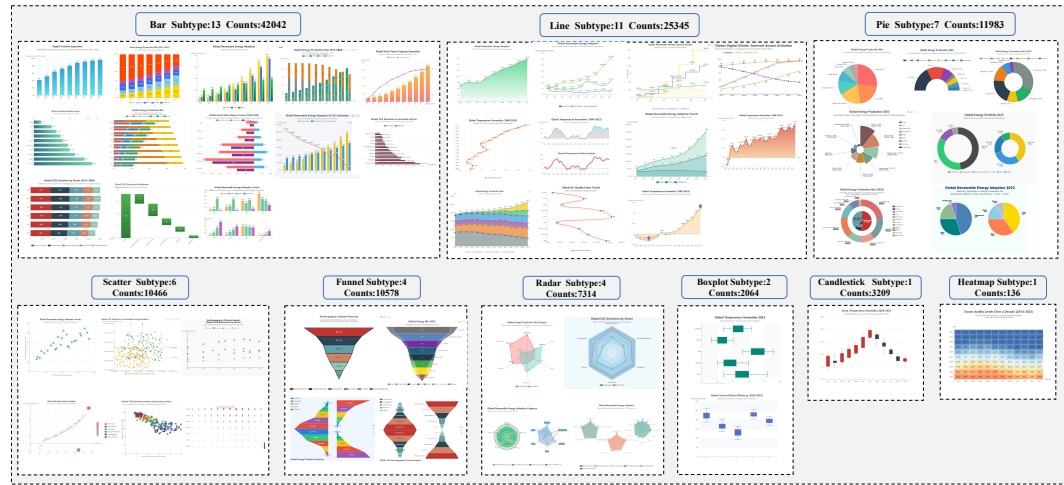

Figure 2: Statistics and distribution of chart types and subtypes in the Chart2Code dataset.

phases: supervised fine-tuning to establish logical competence, followed by reinforcement learning with rule-guided rewards to deepen reasoning capabilities. Overall, our framework treats code as a compositional and interpretable bridge between vision and language, enabling accurate and logically grounded chart understanding, as illustrated in Figure 1.

### 3.1 CHART2CODE

**Echarts-format Chart Generation.** We begin with a template library $\mathcal{T} = \{T_1, T_2, \ldots, T_K\}$, which spans $K$ chart templates across 9 major categories and 49 subtypes. For each template $T_k \in \mathcal{T}$, we construct a prompt $p_k$ to guide an LLM in generating diverse ECharts code. The detailed prompt design is provided in Appendix E. The ECharts code $c_j$ for a sample $j$ is obtained as:

$$c_j = G_{\text{LLM}}(p_k), \tag{1}$$

where $G_{\text{LLM}}(\cdot)$ denotes the LLM that produces structured chart code given the input prompt.

**Quality Filtering Pipeline.** The generated ECharts code is rendered into images and subjected to a rigorous quality control process. We combine automated pixel-level filtering with manual review to enhance image quality. In the automated stage, each image is converted to the Hue-Saturation-Value color space to extract saturation and brightness features and is downsampled to reduce computational overhead. Blank and noisy images are then removed using sparse content detection and white-background noise filtering. In the manual stage, we further eliminate edge cases that are difficult to detect automatically. As a result, we retain approximately 110K high-quality charts from the initial set. Detailed distribution statistics are provided in Figure 2, covering 9 major categories and 49 subcategories. Specific data examples are included in the Appendix F.

**Chart2Code Model.** To enable high-fidelity chart reconstruction from images, we construct a large-scale chart2code dataset and use it to train a multimodal model capable of translating chart images into their corresponding ECharts code. We fine-tune a vision-language model on this dataset for the chart-to-code generation task. Given a chart image $x_i$, the model predicts its corresponding ECharts code sequence $\mathbf{c_i} = (c_{i,1}, c_{i,2}, \ldots, c_{i,L_i})$, where $L_i$ is the token length of the code. The model is trained to maximize the likelihood of the target sequence conditioned on the input image. The model parameters are denoted as $\theta = \theta_{\text{VE}}, \theta_{\text{LD}}$, where $\theta_{\text{VE}}$ refers to the visual encoder (frozen during training), and $\theta_{\text{LD}}$ denotes the language decoder parameters. The training objective is to minimize the loss function $\mathcal{L}_{\text{C2C}}$:

$$\mathcal{L}_{\text{C2C}}(\theta_{\text{LD}}) = -\sum_{i=1}^{N_{\text{C2C}}} \sum_{t=1}^{L_i} \log P(c_{i,t} \mid x_i, \mathbf{c}_{i,<t}; \theta) \tag{2}$$

where $\mathbf{c}_{i,<t}$ represents the sequence of previously generated (ground-truth) tokens $(c_{i,1}, \ldots, c_{i,t-1})$ for the $i$-th sample, $N_{\text{C2C}}$ denotes the total number of samples in the chart2code dataset.

This training strategy enables the model to effectively extract both structural and semantic information from visual inputs and generate accurate, executable code.

## 3.2 CHARTTHINK CONSTRUCTION AND COLLECTION

Current Chart QA datasets primarily consist of image-question-answer triplets, lacking explicit annotations of intermediate reasoning steps. This limits their effectiveness in training models that require step-by-step reasoning grounded in chart content. To address this limitation, we construct a code-driven reasoning dataset that extends traditional QA data with model-generated reasoning paths anchored in chart code. The construction pipeline is as follows.

**ChartThink Construction.** We begin by consolidating existing datasets into a unified collection, denoted as $\mathcal{D}_{\text{orig}} = \{(x_k, q_k, a_k)\}_k$, where $x_k$ represents a chart image, $q_k$ is a corresponding question, and $a_k$ is the ground-truth answer. Each question is annotated with a reasoning type, and each chart is labeled with a structural type. To ensure broad coverage and balanced representation, we perform stratified sampling across both dimensions to select a representative subset. For each sampled chart image $x_k$, we apply the trained Chart2Code model to generate its corresponding ECharts specification $c_k$. The generated code, together with the question $q_k$, is then provided as input to a long-chain reasoning LLM, which outputs a reasoning path $r_k$ and a predicted answer $\tilde{a}_k$:

$$(r_k, \tilde{a}_k) = G_{\text{LC-R}}(\text{Prompt}(\text{Chart2Code}(x_k), q_k)) \tag{3}$$

To ensure data quality, we retain only those samples where the predicted answer $\tilde{a}_k$ exactly matches the ground-truth answer $a_k$. The final constructed dataset, referred to as ChartThink, is defined as:

$$\mathcal{D} = \{(x_j, q_j, r_j, a_j)\}_{j=1}^N \tag{4}$$

Here, $x_j, q_j, r_j, a_j$ represent the chart image, the corresponding question, the generated reasoning path, and the verified answer for the $j$-th sample, respectively. During training, the input to the reasoning model consists of the chart-question pair $(x_j, q_j)$, while the target output is the concatenated sequence of the reasoning path $r_j$ followed by the final answer $a_j$.

**ChartThink Collection.** We construct the ChartThink dataset by aggregating and cleaning a wide range of existing ChartQA datasets, including ChartQA (Masry et al., 2022), EvoChart (Huang et al., 2025a), ChartBench (Xu et al., 2023), and PlotQA (Methani et al., 2020). These datasets collectively encompass diverse chart types and question styles commonly found in practical applications. Following the unified code-driven data pipeline introduced earlier, we systematically process all collected data to ensure consistency and correctness. After filtering out low-quality or mismatched samples, we obtain a high-quality subset containing over 140K examples, each paired with verified answers and intermediate reasoning traces. To better understand the dataset composition, we conduct a detailed analysis of the reasoning types and chart structures. Figure 3 presents the ChartThink dataset statistics and its distribution over four reasoning categories and seven chart types, providing a strong foundation for training models on complex ChartQA.

## 3.3 CHARTREASONER

**Supervised Fine-Tuning.** The reasoning model is first trained using SFT on the $\mathcal{D}$ dataset. Given a chart image $x_j$ and a question $q_j$, the model is trained to generate a target output sequence $\mathbf{y}_j = (y_{j,1}, y_{j,2}, \ldots, y_{j,K_j})$, which consists of a reasoning path followed by the final answer, and contains $K_j$ tokens. The model parameters are denoted as $\theta = \{\theta_{\text{VE}}, \theta_{\text{LD}}\}$, where $\theta_{\text{VE}}$ refers to the visual encoder (kept frozen during training), and $\theta_{\text{LD}}$ denotes the parameters of the language decoder. The SFT objective is to minimize the loss function $\mathcal{L}_{\text{SFT}}$, is defined as:

$$\mathcal{L}_{\text{SFT}}(\theta_{\text{LD}}) = -\sum_{j=1}^N \sum_{t=1}^{K_j} \log P(y_{j,t} \mid x_j, q_j, \mathbf{y}_{j,<t}; \theta) \tag{5}$$

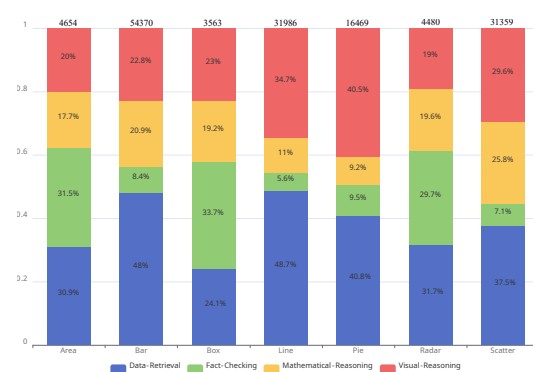

Figure 3: ChartThink dataset statistics and distribution.

where $\mathbf{y}_{j,<t}$ represents the sequence of previously generated (ground-truth) tokens $(y_{j,1}, \ldots, y_{j,t-1})$ for the $j$-th sample.

This approach improves response uniformity and provides a stable foundation for the subsequent reinforcement learning stage.

**Reinforcement Learning with GRPO.** While supervised fine-tuning equips the model with fundamental chart understanding, it also reveals a common failure mode: over-generation of verbose reasoning chains, even when the input lacks sufficient information. This over-reasoning behavior compromises answer reliability. To mitigate this, we adopt a reinforcement learning phase using GRPO. Unlike standard policy optimization methods such as PPO (Schulman et al., 2017), GRPO generates multiple candidate responses per input and optimizes them jointly via intra-group normalization. This stabilizes training and encourages the model to favor concise and accurate outputs.

We design structured, rule-based reward functions that explicitly measure answer quality across dimensions such as factual accuracy, formatting correctness, and response length. These reward signals guide the model to suppress hallucinations and over-reasoning, promoting disciplined and generalizable reasoning. Overall, this RL phase aligns the model's outputs with practical expectations and user preferences, significantly enhancing robustness across diverse ChartQA scenarios.

## 4 EXPERIMENTS

### 4.1 EXPERIMENTAL SETUP

**Datasets.** We evaluate ChartReasoner on four benchmarks—ChartQA (Masry et al., 2022), EvoChart-QA (Huang et al., 2025a), ChartQAPro (Masry et al., 2025a), and ChartBench (Xu et al., 2023)—covering diverse chart types and reasoning skills, from simple plots to real-world dashboards and multi-chart compositions. EvoChart-QA and ChartQAPro serve as out-of-domain tests for generalization to realistic, structurally diverse charts; ChartQAPro also includes multi-turn, hypothetical, and unanswerable queries. ChartBench provides large-scale, type-diverse evaluation. We further assess Chart2Code on EvoChart-QA to measure chart reconstruction. Additional setup details are in Appendix B.

**Evaluation Metrics & Baselines.** For ChartQA, we follow the official protocol for each benchmark. For Chart2Code, we adopt execution success rate and GPT-4V [1] visual similarity scoring (1–10), following Plot2Code (Wu et al., 2025). The specific prompt is provided in Appendix D.

We benchmark our model against a wide range of MLLMs, including proprietary models like Claude-3.5-Sonnet[2], Gemini-Flash-1.5/2.0 (Team et al., 2024), GPT-4-turbo, and GPT-4o (Achiam et al., 2023), as well as open-source models such as InternVL2 (Chen et al., 2024b), Phi-3-Vision (Abdin et al., 2024), LLaVA-V1.5 (Liu et al., 2024), InternLM-XComposer (Dong et al., 2024), Qwen-VL (Bai et al., 2025; Wang et al., 2024), and CogVLM2 (Hong et al., 2024). We also

---

[1]The version is gpt-4-vision-preview

[2]https://www.anthropic.com/news/claude-3-5-sonnet

| Model Name | Size(B) | Evochart-QA | ChartQA | ChartBench | ChartQAPro |
|---|---|---|---|---|---|
| **Closed-source** | | | | | |
| Claude-3.5-Sonnet | – | 56.96 | **90.80** | 56.95 | 43.58 |
| Gemini-2.0-Flash (Team et al., 2024) | – | **64.64** | 84.80 | 55.63 | **46.85** |
| Gemini-1.5-Flash (Team et al., 2024) | – | 27.90 | 79.00 | 51.42 | 42.96 |
| Gemini-1.5-Pro (Team et al., 2024) | – | 32.20 | 87.20 | 56.26 | 43.24 |
| GPT-4-turbo (Achiam et al., 2023) | – | 40.30 | 62.30 | 43.14 | 32.14 |
| GPT-4o (Achiam et al., 2023) | – | 49.80 | 85.70 | **59.45** | 37.67 |
| **Open-source** | | | | | |
| Intern-VL2.5 (Chen et al., 2024a) | 78 | **57.44** | **88.30** | **65.57** | **45.31** |
| QvQ-Preview (Bai et al., 2025) | 72 | 54.32 | 86.48 | 42.40 | 41.26 |
| Qwen2-VL (Wang et al., 2024) | 72 | 54.00 | **88.30** | 54.77 | 39.42 |
| Intern-VL2 (Chen et al., 2024b) | 40 | 49.00 | 86.20 | 39.63 | 35.21 |
| CogVLM2 (Hong et al., 2024) | 19 | 21.90 | 81.00 | 35.14 | 31.47 |
| Intern-VL2 (Chen et al., 2024b) | 8 | 38.60 | 81.50 | 37.80 | 22.53 |
| Intern-VL2.5 (Chen et al., 2024a) | 8 | 45.20 | 84.80 | 52.96 | 35.67 |
| LLaVA-v1.5 (Liu et al., 2024) | 7 | 17.40 | 55.32 | 23.39 | 16.33 |
| Internlm-XComp.-v2 (Dong et al., 2024) | 7 | 27.80 | 72.64 | 47.78 | 23.75 |
| QwenVL-Chat (Bai et al., 2023) | 7 | 19.70 | 83.00 | 26.98 | 35.59 |
| Qwen2.5-VL (Bai et al., 2025) | 7 | 46.80 | 85.00 | 54.06 | 36.61 |
| Phi3-Vision (Abdin et al., 2024) | 4 | 39.50 | 81.40 | 35.06 | 24.73 |
| **Chart MLLMs** | | | | | |
| ChartLlama (Han et al., 2023) | 13 | 9.50 | 69.66 | 21.71 | 7.28 |
| ChartAst-S (Meng et al., 2024) | 13 | 12.90 | 79.90 | 11.27 | 8.15 |
| ChartIns-Llama2 (Masry et al., 2024) | 7 | 16.80 | 66.64 | 25.28 | 4.88 |
| EvoChart (Huang et al., 2025a) | 4 | **54.20** | 81.50 | 36.14 | 25.05 |
| ChartIns-FlanT5 (Masry et al., 2024) | 3 | 24.30 | 64.20 | 22.15 | 4.27 |
| ChartGemma (Masry et al., 2025b) | 3 | 30.60 | 80.16 | 35.24 | 6.84 |
| TinyChart (Abdin et al., 2024) | 3 | 25.50 | 83.60 | 28.86 | 13.25 |
| ChartReasoner-SFT(Ours) | 7 | 47.04 | 86.76 | 55.10 | 37.94 |
| ChartReasoner-GRPO(Ours) | 7 | 48.10 | **86.93** | **55.20** | **39.97** |

Table 1: Comparisons of ChartReasoner and Baselines on Four ChartQA Benchmarks.

include domain-specific baselines: ChartLlama (Han et al., 2023), ChartAst (Meng et al., 2024), ChartIns (Masry et al., 2024), ChartGemma (Masry et al., 2025b), TinyChart (Zhang et al., 2024), and EvoChart (Huang et al., 2025a). For Chart-to-Code, we use ChartCoder (Zhao et al., 2025) as the main baseline. To assess the structural richness of our dataset, we perform controlled training using Qwen2.5-VL-7B (Bai et al., 2025) on both EvoChart and our dataset.

## 4.2 MAIN RESULTS

**ChartQA Results.** We comprehensively evaluate our ChartReasoner model and a wide range of baseline models, including both general-purpose MLLMs and chart-specialized models, across four benchmark datasets. Among them, ChartQA and ChartBench are in-domain datasets, while ChartQAPro and EvoChart-QA serve as out-of-domain evaluations to test generalization performance. The results are shown in Table 1.

In the ChartQA benchmark, the proprietary Claude-3.5-Sonnet model achieves top-tier performance. However, our ChartReasoner significantly outperforms all open-source 7B models and surpasses the majority of chart-specialized baselines, demonstrating its strong reasoning capability in structured visual tasks. A similar trend is observed in ChartBench, where GPT-4o leads among proprietary models, yet our model achieves state-of-the-art results among open-source and domain-specific competitors. These findings confirm that while proprietary models still retain an edge on in-domain datasets, strengthening reasoning and analysis ability can bridge this gap and yield competitive results. In the EvoChart-QA benchmark, which contains long and complex real-world charts, GPT-4o shows relatively weaker performance. In contrast, EvoChart, a chart-specialized model trained on similar data, performs better but shows clear limitations on ChartQA, indicating limited cross-domain generalization due to data-specific overfitting and a smaller model scale. Notably, our ChartReasoner matches GPT-4o's performance and outperforms its own base model Qwen2.5-VL, confirming its enhanced capacity for long-chain visual reasoning and data adaptation. Lastly, in the ChartQAPro benchmark, Gemini-Flash-2.0 stands out among proprietary models. Still, ChartReasoner surpasses even GPT-4o in this domain-shifted setting. This reveals that many proprietary models struggle with domain transfer in chart understanding, whereas ChartReasoner's consistent

performance under both in-domain and out-of-domain conditions underscores the importance of improving reasoning and abstraction abilities to enhance chart-centric generalization.

Models refined using GRPO after SFT consistently outperform those trained with SFT alone across all evaluation benchmarks. In addition to improvements in accuracy, GRPO significantly enhances the quality of reasoning by minimizing excessive explanations and fostering outputs that are more structured and precise. Our further analysis on the ChartQA test set reveals that GRPO reduces the length of excessively long reasoning chains, with the average token length decreasing from 699.03 to 618.22. Moreover, GRPO completely eliminates the issue of truncated responses, a problem observed in the SFT model, where 59 instances of truncation occurred, compared to zero with GRPO. These results demonstrate the dual benefits of GRPO: improving overall performance while optimizing the reasoning process, enabling the model to generate more concise, logically consistent, and reliable outputs, particularly in visual reasoning tasks.

### 4.3 ABLATION STUDY

**Chart-to-Code Performance Evaluation.**  We report comprehensive results in Table 2, aggregating comparisons across datasets and training scales. For a real-world EvoChart-QA test set, Chart2Code is evaluated using GPT-4V visual-similarity scores and pass rates.

Our Chart2Code, trained on the proposed ECharts-based dataset, significantly outperforms models trained on the EvoChart dataset under comparable training sizes. The results highlight notable improvements in both visual fidelity and pass rate, demonstrating the higher quality and diversity of our data. This suggests that our dataset enables better generalization and more accurate chart reconstruction, even for complex and diverse chart types encountered in practice.

In addition, our model trained on ECharts-based data exhibits superior performance compared to those trained on large-scale, Python-generated chart datasets. Despite the latter having access to more training examples, their performance lags behind in both robustness and reconstruction accuracy. This underscores the importance of data realism and expressiveness, qualities more inherently present in ECharts specifications, for effectively training chart generation models.

We also analyze the impact of data volume by training models on subsets of 30k, 50k, 70k, and 110k chart–code pairs. Results show that while increasing the dataset size generally improves performance, the gains begin to plateau beyond 70k examples. This saturation effect suggests that 110k samples are sufficient to maximize the model's reconstruction capability.

| Model | Data | Similarity | bar | line | pie | scatter | Rate | Types |
|-------|------|-----------|-----|------|-----|---------|------|-------|
| ChartCoder | 160k | 3.64 | 4.18 | 3.91 | 3.25 | 3.22 | 82.40% | 27 |
| Chart2Code-Evo. | 70k | 3.84 | 4.63 | 4.16 | 3.94 | 2.63 | 89.10% | 4 |
| Chart2Code | 30k | 2.39 | 3.12 | 2.24 | 2.81 | 1.39 | 88.20% | 49 |
| Chart2Code | 50k | 3.62 | 4.37 | 3.81 | 4.17 | 2.13 | 90.60% | 49 |
| Chart2Code | 70k | 4.21 | 5.17 | 4.20 | 4.23 | 3.24 | 91.00% | 49 |
| Chart2Code | 110k | **4.34** | **5.26** | **4.21** | **5.12** | **3.77** | **92.40%** | 49 |

Table 2: Chart2Code Performance on EvoChart-QA: GPT-4V Similarity Scores (including Breakdown by Chart Types) and Overall Pass Rates.

**Sensitivity Analysis**  We conduct a sensitivity analysis to evaluate how chart type affects both chart reconstruction and downstream reasoning performance. As shown in Table 2, the Chart2Code module exhibits strong performance on bar and pie charts, while its accuracy declines for scatter and line charts. Scatter plots often contain dense, overlapping points that hinder precise encoding, whereas many line charts in EvoChart are multi-series or include complex visual encodings, making them particularly challenging to parse. These characteristics, along with their relative scarcity in the training data, contribute to consistently lower reconstruction accuracy, especially for line charts, across all models.

This reconstruction quality directly influences reasoning performance in the ChartReasoner module. As shown in Figure 4 and Figure 5, ChartReasoner achieves competitive results on bar and pie charts but underperforms on scatter, line, and box plots. Notably, the stronger results on bar, line, and pie charts within ChartBench align with their higher frequency in our training data, which enhances reconstruction robustness and, in turn, improves reasoning accuracy. These observations highlight

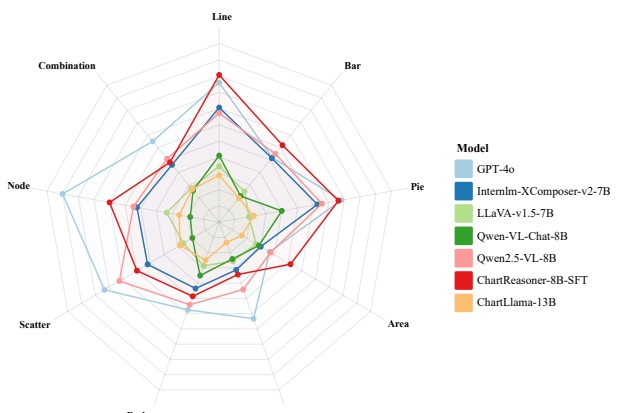

Figure 4: ChartBench Performance Across Chart Types.

| DataSet | Evochart-QA | ChartQA | ChartBench | ChartQAPro |
|---|---|---|---|---|
| ChartQA | 41.3 | **86.56** | 51.43 | 35.64 |
| EvoChart | **42.8** | 85.48 | 52.38 | **36.05** |
| ChartBench | 40.5 | 81.56 | **54.76** | 32.36 |
| PlotQA | 40.2 | 83.00 | 47.80 | 34.69 |

Table 3: Impact of Different Dataset Sources on Downstream Chart Reasoning Performance.

a strong correlation between reconstruction reliability and downstream performance, underscoring the importance of both visual complexity and data distribution in ChartQA tasks.

**Impact of Different Dataset Sources.** We further investigate how the choice of training datasets influences downstream chart reasoning performance when generating chain-of-thought examples. To this end, we sample 20k instances from ChartQA, EvoChart, ChartBench, and PlotQA, and convert them into reasoning examples using our chart-to-code distillation pipeline. As shown in Table 3, using training data that share the same distribution as the evaluation benchmark consistently leads to the best performance, highlighting the importance of distribution alignment. Among the evaluated datasets, PlotQA yields the lowest performance across all benchmarks. This outcome is likely due to its synthetic construction, limited visual variety, and narrow set of chart types, which are primarily restricted to bar, line, and dot charts. These characteristics make it less representative of real-world chart scenarios. In comparison, training data derived from EvoChart achieve better generalization, especially on EvoChart-QA and ChartQAPro. EvoChart includes a broader range of chart types, such as pie and scatter, and its charts are more visually aligned with those found in practical applications, which improves cross-domain performance. ChartBench offers strong results when evaluated on its own distribution but shows reduced effectiveness on other benchmarks, suggesting limited transferability. Overall, these findings emphasize the importance of dataset diversity when training chart reasoning models capable of robust generalization.

## 5 CONCLUSION

We present ChartReasoner, a code-driven two-stage framework that bridges visual chart understanding and long-chain reasoning in multimodal LLMs. Our approach introduces Chart2Code, which converts chart images into high-fidelity ECharts code, and builds ChartThink, a dataset of 140K+ multi-step reasoning samples, which enables precise, interpretable, and scalable ChartQA. Unlike prior methods that rely on shallow CoT or lossy image-to-text conversion, ChartReasoner uses structured symbolic representations to preserve layout, semantics, and quantitative details. Extensive evaluations on four benchmarks show strong generalization and reasoning performance. The framework supports faithful information extraction and logical inference, yielding more accurate and transparent decisions. Overall, our results highlight the value of symbolic representation and structured reasoning by tightly integrating visual parsing with logical inference.

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

## A   APPENDIX

## B   IMPLEMENTATION DETAILS.

We use DeepSeek-R1 as a controllable code and reasoning generator in both stages of data construction. We use Qwen2.5-VL-7B (Bai et al., 2025) as the backbone and perform supervised fine-tuning on 8 A100 80GB GPUs. The vision tower and projection layers are frozen, while the language model is fully trainable. Training runs for 4 epochs with an effective batch size of 8, using BF16 precision. We apply the AdamW (Loshchilov & Hutter, 2019) optimizer with learning rate 1e-5. The maximum sequence length is 4096 tokens, and images are resized to 512×512 pixels. We further apply GRPO for 2 epochs starting from the SFT checkpoint. The model generates 8 completions per input, with reward-weighted selection based on accuracy, format correctness, and length suitability.

## C   QUALITATIVE ANALYSIS

To further illustrate the performance improvements brought by our model in chart-based multimodal reasoning, we conduct a qualitative analysis using representative examples. These cases help

demonstrate how enhanced reasoning capabilities can effectively assist visual understanding, especially when direct visual recognition is ambiguous or when the question requires complex logical interpretation. As illustrated in Figure 6–Figure 9, these examples further demonstrate the effectiveness of our method.

**Visual-Aided Reasoning.** One core strength of our ChartReasoner lies in its ability to perform visual reasoning that supplements and corrects potentially uncertain visual recognition. As shown in Figure 6, the example question is: "What is the label of the highest bar of February?" This task requires the model to first locate February on the x-axis and then identify the label corresponding to its highest bar—thus constituting a visual reasoning problem.

While baseline models such as Qwen2.5VL fail to correctly locate "February" and incorrectly identify "Sales" as the highest category, ChartReasoner demonstrates a more accurate analysis by first reasoning through the axis structure: "The x-axis data is [January, February, March, ..., December], so February is the second month." This allows it to correctly localize the February column and extract the corresponding bar label, thereby arriving at the correct answer.

This example highlights that reasoning capabilities can effectively compensate for limitations in visual recognition, particularly when axis elements or data labels are densely packed, occluded, or ambiguously rendered.

**Complex Semantic Reasoning.** In addition to visual grounding, ChartReasoner also excels in handling complex semantic questions that require precise logical understanding. As shown in Figure 7, the example question is: "How many percent of U.S. coffee drinkers drink less than 2 cups of coffee at home on a weekday?" The key to this question lies in correctly interpreting the condition "less than 2 cups." However, Qwen2.5VL incorrectly includes the "2 cups" category in its calculation, leading to a wrong answer. In contrast, ChartReasoner demonstrates its advanced reasoning by recognizing the logical boundary of the query and explicitly excluding the 2-cup group from its aggregation, yielding the correct answer. This indicates that reasoning ability is critical for precise comprehension of quantitative and conditional logic, which is often required in real-world ChartQA scenarios.

## D    PROMPT DESIGN FOR VISUAL EVALUATION WITH GPT-4V

To comprehensively assess the visual quality of generated charts, we adopt a structured prompt-based evaluation approach using GPT-4V. The prompt instructs the model to compare a generated chart with its corresponding ground-truth version and assign a similarity score ranging from 1 to 10. The scoring is based on four key criteria: Colors (accuracy of color schemes), Axes & Scale (consistency of axis ranges and units), Data Points Position (placement and alignment of bars, lines, or markers), and Overall Layout (correctness of titles, labels, legends, etc.).

This prompt enables GPT-4V to produce fine-grained visual judgments that go beyond traditional execution-based metrics (e.g., code correctness), capturing layout-level discrepancies that impact real-world interpretability. An example of such a prompt is illustrated in Figure 10. This evaluation method bridges the gap between syntactic correctness and perceptual fidelity in chart generation tasks.

## E    PROMPT ENGINEERING FOR ECHARTS CODE GENERATION

To enable effective chart generation, we employ domain-specific prompt engineering tailored to the ECharts visualization framework. The prompts are constructed to cover 18 thematic domains and 111 subtopics, spanning social, economic, technological, and environmental dimensions. This ensures diverse coverage of chart types and semantic contexts.

Each prompt clearly specifies the chart topic, the intended visual form (e.g., bar chart, line chart, scatter plot), and any constraints on the layout or data encoding. As demonstrated in Figure 11, this guided prompting allows models like DeepSeek R1 to leverage their strong reasoning abilities to produce structurally varied and semantically rich visualizations. These prompts are essential to

ensure that the generated charts are not only syntactically valid but also meaningful and domain-relevant.

## F    CHART2CODE DATASET DETAILED CASE

To further illustrate the design of our Chart2Code Dataset, we present selected examples that directly show the generated ECharts HTML code alongside the corresponding rendered chart. These examples also highlight the flexibility of the chart template system and the reasoning capability of the DeepSeek R1 model in generating structurally complex and thematically rich charts. By showcasing a range of chart types—including bar, line, and pie charts—these cases reflect the robustness of our prompt engineering approach and the effectiveness of the multi-stage quality filtering pipeline described in the methodology. Figure 12–Figure 15 present more detailed examples from the Chart2Code dataset.

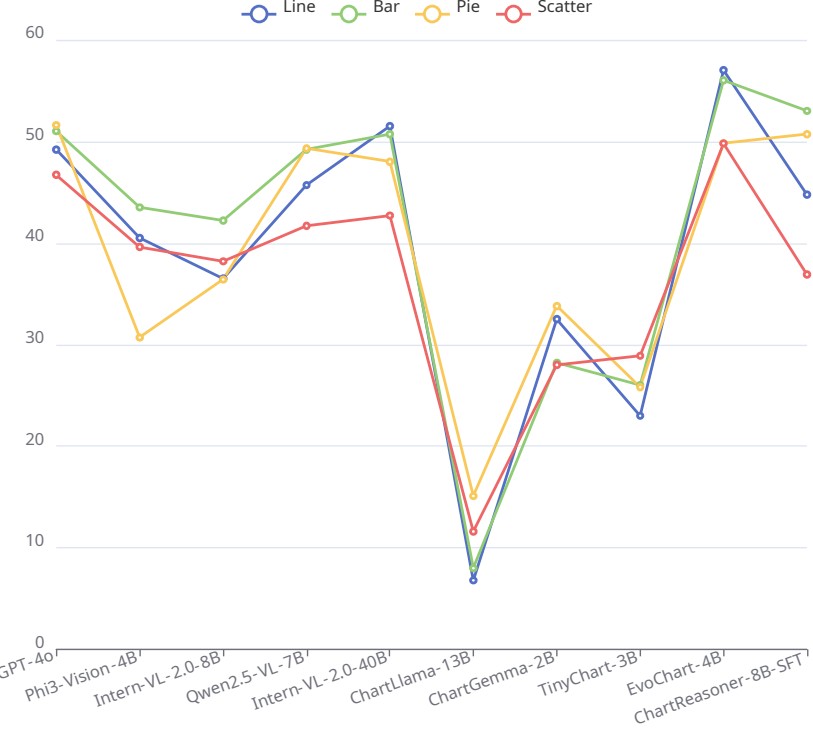

Figure 5: EvoChart Performance Across Chart Types.

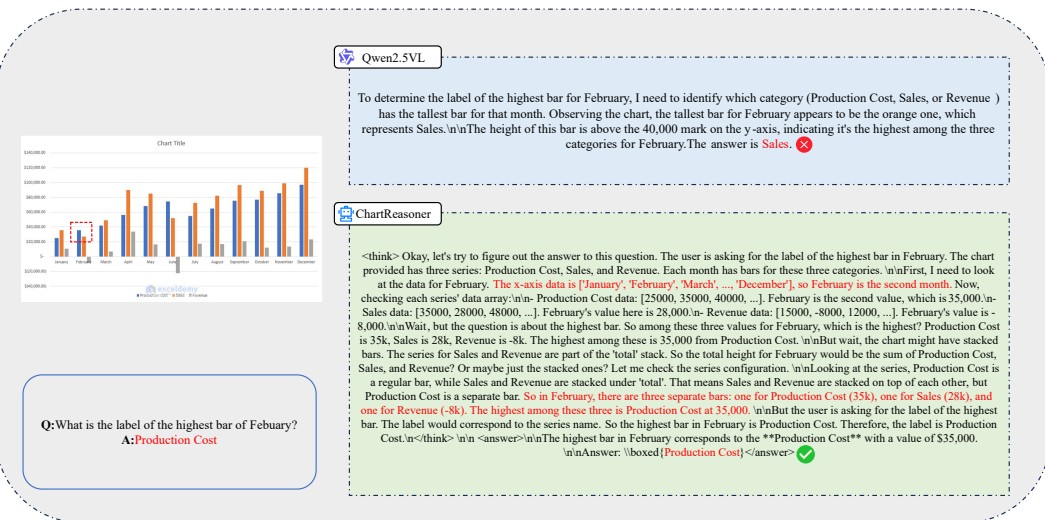

Figure 6: Comparison of Model Responses in ChartQA on a Bar Chart.

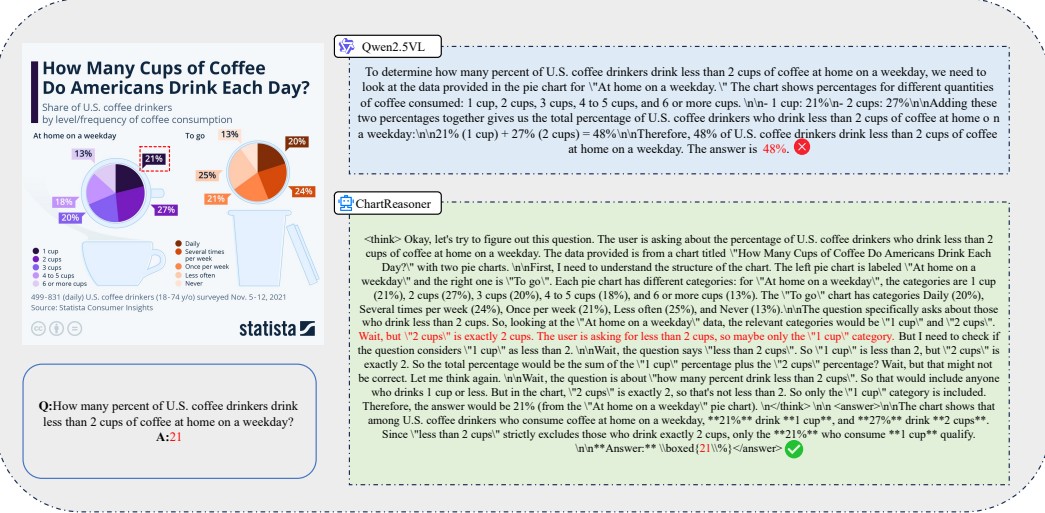

Figure 7: Comparison of Model Responses in ChartQA on a Pie Chart.

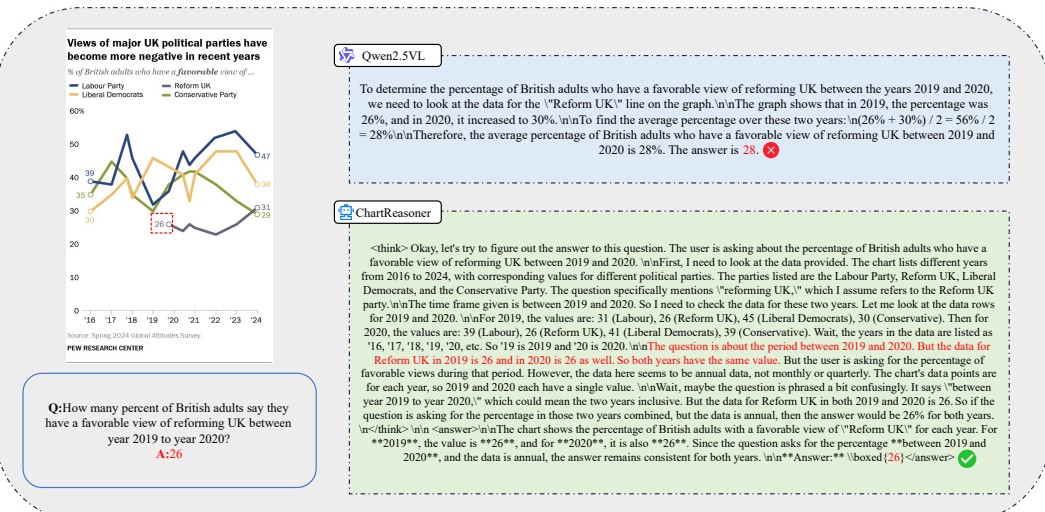

Figure 8: Comparison of Model Responses in ChartQA on a Line Chart.

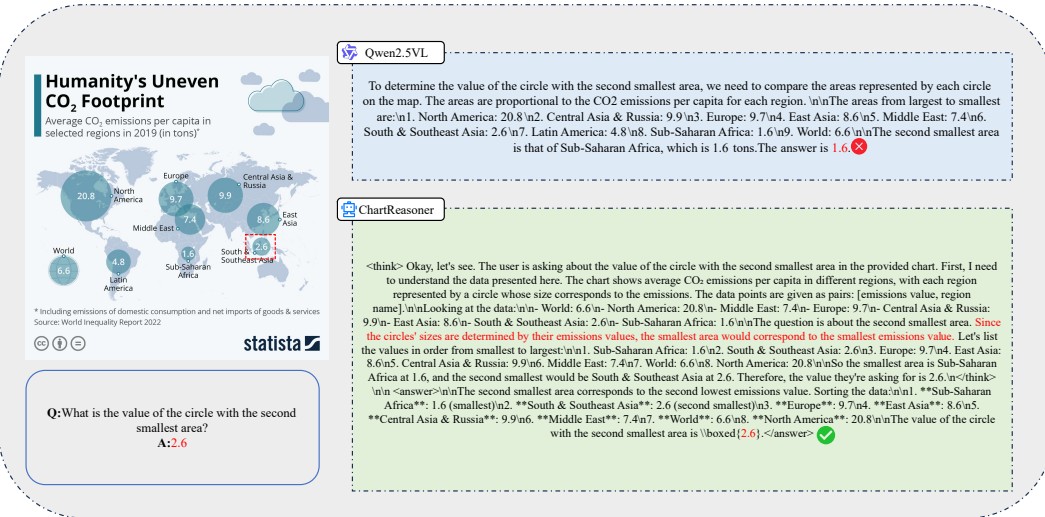

Figure 9: Comparison of Model Responses in ChartQA on a Scatter Chart.

## Prompt (GPT-4V Visual Evaluation.)

Please evaluate the similarity between a reference image created using matplotlib and an image generated by code provided by an AI assistant. Consider factors such as the overall appearance, colors, shapes, positions, and other visual elements of the images. Begin your evaluation by providing a short explanation. Be as objective as possible. After providing your explanation, you must rate the response on a scale of 1 to 10 by strictly following this format: Rating: [[5]]

Figure 10: GPT-4V Visual Evaluation Prompt.

918
919
920
921
922
923
924

**Prompt (ECharts Code Generation.)**

925
926

You are a web chart generation assistant. Please emulate the structure, style, and configuration of the ECharts chart in the following HTML: (Original chart HTML below)

927

{echarts_template}

Modify it according to these requirements and generate a complete HTML page with the chart (ready to run in a browser). Choose one theme and dataset from the list below, or combine multiple for richer context, but feel free to create your own variation:

928
929
930

Pay attention to keep the data distribution reasonable and diverse when drawing the graph, consider the rendering effect, and conform to the real chart

Note that the scatter plot distribution is random and should not be concentrated together

931

Climate and Environment: global temperature anomalies, CO2 emissions by sector, deforestation rates, sea level rise, ocean acidity, renewable energy adoption, air quality index, water scarcity index, glacier retreat

932
933

Population and Demographics: world population growth, urban vs rural distribution, age pyramids by country, migration patterns, household income distribution, gender ratio statistics, life expectancy trends

934
935

Economics and Finance: stock market indices (e.g., SandP 500, FTSE 100), GDP per capita, inflation rates, foreign direct investment, income inequality (Gini coefficient), cryptocurrency market capitalization, commodity prices (oil, gold, agriculture)

936
937

Energy and Resources: solar and wind power capacity, oil and gas production, nuclear energy share, water consumption per capita, mineral extraction volumes, waste recycling rates, renewable vs non-renewable energy mix

938

Technology and Internet: global internet penetration, mobile phone subscriptions, social media user growth, e-commerce sales, cybersecurity incidents, data center energy usage, AI investments, open source contribution trends

939
940

Health and Society: pandemic case numbers, vaccination rollout rates, healthcare expenditure per capita, mental health survey scores, hospital bed availability, disease incidence rates, life satisfaction index

941

Retail and Sales: monthly retail sales by sector, online vs offline revenue, average basket size, foot traffic in malls, customer churn rate, loyalty program engagement

942
943

Education and Employment: enrollment rates in primary/secondary/tertiary, literacy rates, graduation rates by discipline, job vacancy data, unemployment rates, average salary by industry, remote work adoption, skill shortage indices

944
945

Tourism and Transportation: tourist arrivals by region, airline passenger miles, ride-sharing usage, public transit ridership, port container throughput, traffic congestion index, vehicle electrification adoption

946

Sports and Entertainment: sports league attendance, athlete medal counts, box office revenue by genre, music streaming hours, video game sales figures, award show winners stats, TV viewership ratings

947

Media and Communication: newspaper circulation, podcast listenership, YouTube subscriber growth, mobile app usage time, online news article shares, media trust index

948

Automotive and Mobility: vehicle sales by type (EV, ICE, hybrid), autonomous vehicle tests, public bike-share usage, traffic accident statistics, fuel efficiency trends, ride-hailing market share

949
950

Agriculture and Food: crop yield per hectare, food price index, livestock population, organic farming acreage, seafood harvest volumes, global hunger index

951

Science and Research: scientific publication counts by field, research funding allocation, patent filings, RandD expenditure, Nobel prize distribution, clinical trial numbers

952
953

Real Estate and Construction: housing price index, construction starts by region, mortgage interest rates, commercial real estate vacancies, smart city projects

954

Government and Public Policy: budget deficit/surplus, tax revenue breakdown, public debt levels, policy approval ratings, crime rates by category, election turnout statistics

955

Space and Aeronautics: satellite launches, ISS research hours, space tourism bookings, Mars rover milestones, asteroid detection counts Miscellaneous: cryptocurrency price volatility, earthquake frequency and magnitude, festival attendance, book publication counts, open source project activity

956
957

Use your imagination and knowledge to create different data distributions based on the topic.

Pay attention to keep the data distribution reasonable and diverse when drawing the graph, consider the rendering effect, and conform to the real chart.

958
959

Note that the scatter plot distribution is random and should not be concentrated together

1. Replace the chart data with a different but coherent dataset.

960

2. The data distribution and trends should be as complex as possible and not too monotonous.

961

3. Change the topic or theme accordingly.

962

4. Add the main title and subtitle related to the new topic and let your imagination run wild.

963

5. Keep the original chart type.

6. You can use your imagination to change the style and color at will.

964

7. Return only the full HTML code—no explanations or comments.

965
966
967

Figure 11: ECharts Code Generation Prompt.

968
969
970
971

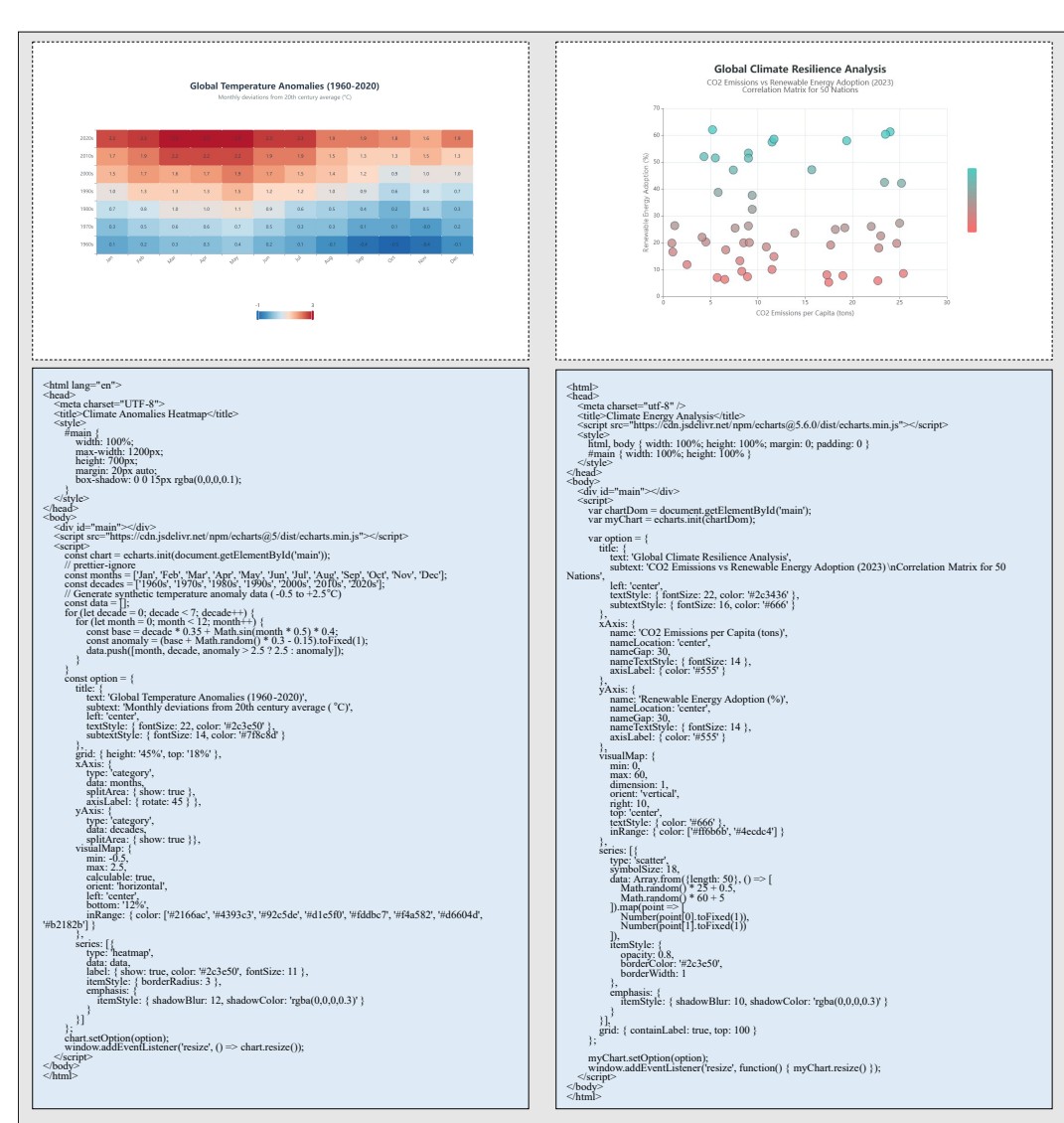

Figure 12: Example 1 from the Chart2Code Dataset.

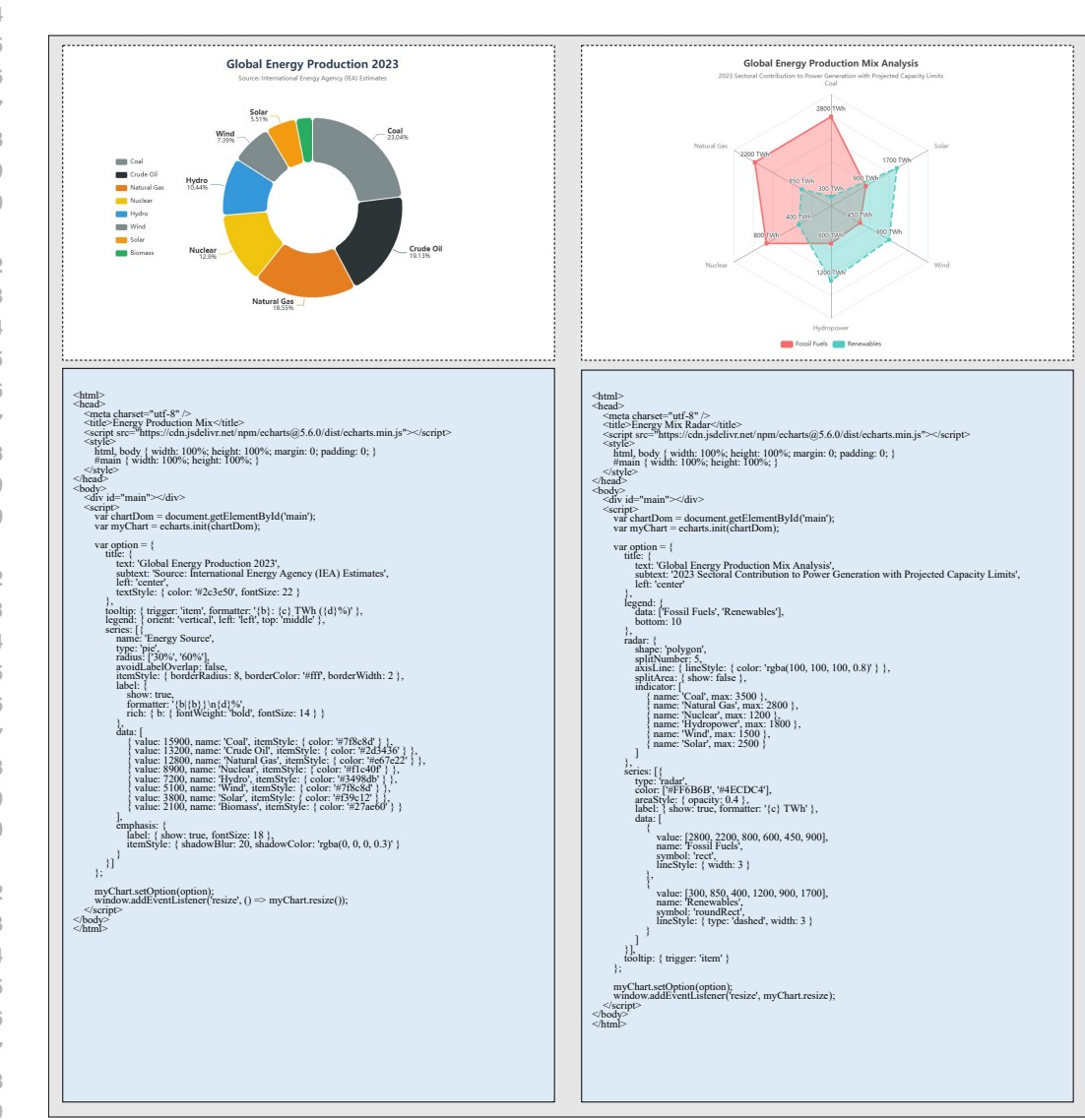

Figure 13: Example 2 from the Chart2Code Dataset.

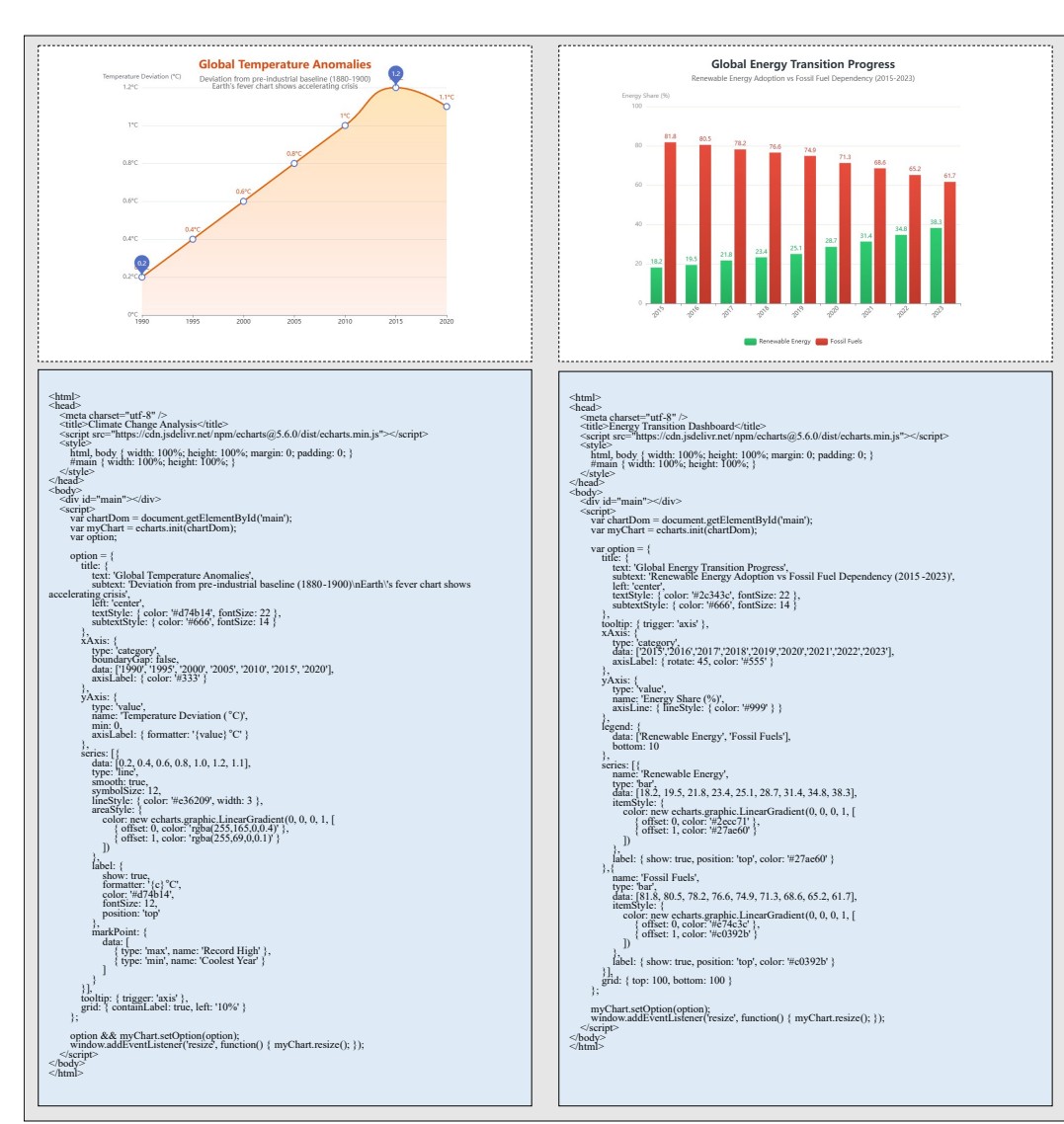

Figure 14: Example 3 from the Chart2Code Dataset.

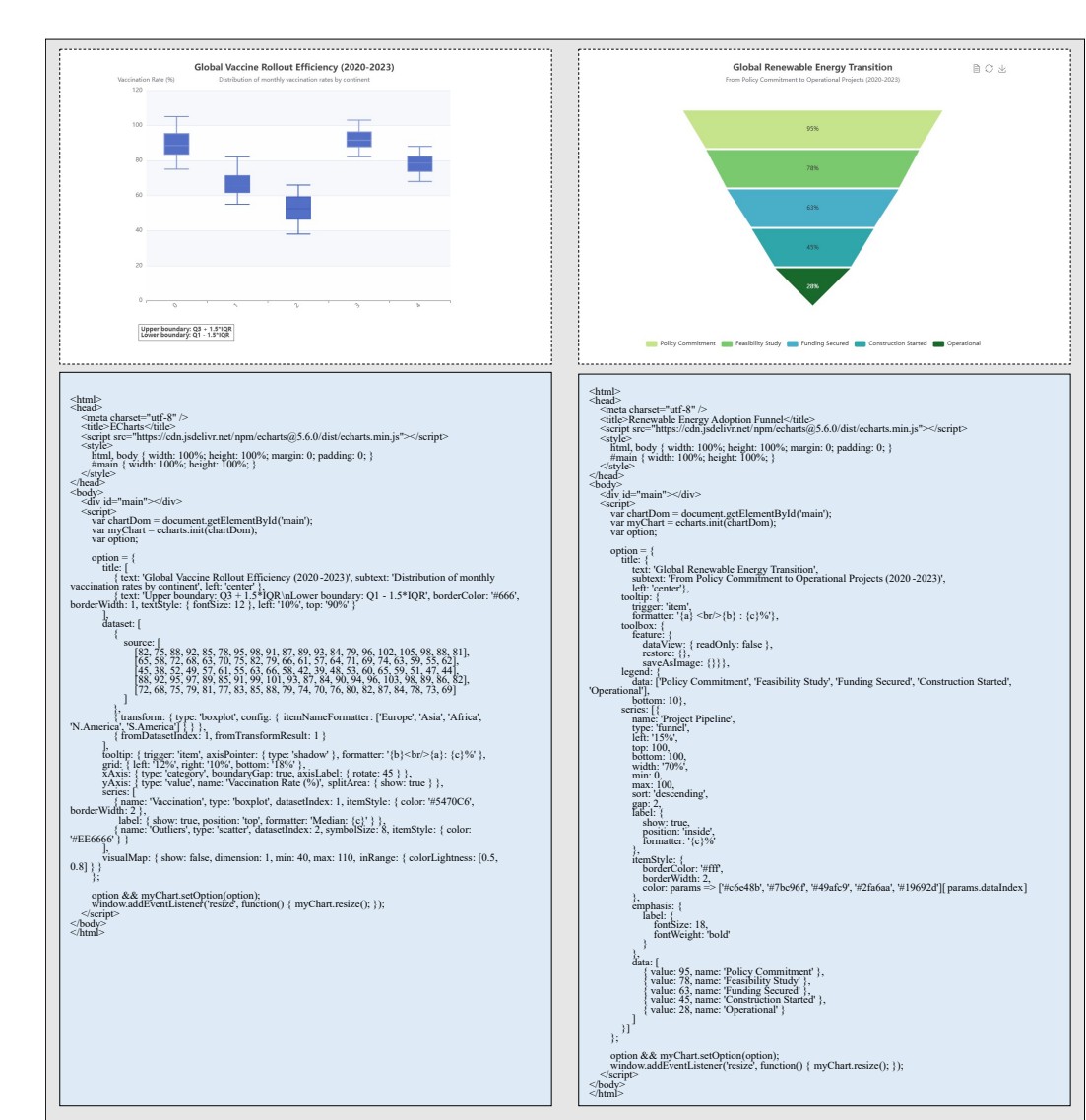

Figure 15: Example 4 from the Chart2Code Dataset.

