# OpenReview forum: "ChartReasoner: Code-Driven Modality Bridging for Long-Chain Reasoning in Chart Question Answering"
_ICLR.cc/2026/Conference — Submitted to ICLR 2026_

### Official Review · Reviewer_UpE9 · 2025-10-17

**Soundness:** 1
**Presentation:** 2
**Contribution:** 2
**Rating:** 2
**Confidence:** 4

**Summary:**

This paper introduces ChartReasoner, a two-stage, code-driven framework designed to enhance long-chain reasoning for MLLMs on Chart Question Answering (ChartQA) tasks. The authors argue that conventional image-to-text methods lose critical information. Their proposed solution first uses a model, Chart2Code, to translate a chart image into a high-fidelity, structured ECharts code representation. This stage is supported by a new synthetic dataset of 110K image-code pairs. In the second stage, this Chart2Code model is applied to existing benchmarks (ChartQA, ChartBench, etc.) to generate code, which is then fed to a long-chain reasoning LLM (DeepSeek-R1) to generate reasoning paths. These paths, filtered by final answer correctness, form a new 140K-sample dataset called ChartThink. The final ChartReasoner model is then trained on this dataset using a combination of Supervised Fine-Tuning (SFT) and Reinforcement Learning (GRPO). The central idea is that code serves as a superior, lossless intermediate modality for complex reasoning.

**Strengths:**

1. Interesting Idea: The core idea of using executable code (ECharts) as a symbolic, intermediate representation to bridge the visual-textual modality gap is interesting.

2. Substantial Dataset Contributions: The paper introduces two large-scale datasets: Chart2Code for image-to-code translation and ChartThink for code-based reasoning. The creation and release of these resources are a significant contribution to the community.

**Weaknesses:**

1. High Risk of Train-Test Overlap: This is the most critical issue. The paper states that the ChartThink training dataset is constructed by processing samples from existing benchmarks, including ChartQA and ChartBench. It then evaluates the final model on the test sets of these same benchmarks, labeling them as "in-domain." The paper provides no clarification on whether it excluded the official test splits of these benchmarks during the creation of its training data. Without this explicit separation, there is a high probability of data contamination, where the model has been trained on samples derived from the test set it is being evaluated on. This potential data leakage makes the reported results on these benchmarks unreliable and possibly invalid.

2. Questionable Necessity of the Intermediate Code Step: Although the idea is interesting, the paper's core premise is conceptually questionable. The ability of the Chart2Code model to generate accurate, detailed ECharts code from an image implies that it has already achieved a deep and structured understanding of the chart's components, layout, and data. If the model already possesses this rich internal representation, the necessity of an explicit, separate code-generation step is unclear. It seems plausible that a model with this level of visual parsing capability could be trained to reason directly on its internal representations, making the two-stage pipeline an unnecessarily complex and potentially inefficient detour. The paper fails to justify why this "extra step" is indispensable.

3. Quality of Generated Reasoning Traces: The reasoning paths in the ChartThink dataset, which form the basis for the final model's training, are generated automatically by prompting an LLM (DeepSeek-R1). The only quality control measure is to filter out samples where the LLM's final answer does not match the ground truth. This is a very weak form of supervision. It does not guarantee that the reasoning path itself is correct, logical, human-like, or even the most efficient way to solve the problem. The final ChartReasoner model is therefore trained via imitation learning on potentially flawed, unnatural, or suboptimal reasoning logic, which limits the quality and robustness of what it can learn.

**Questions:**

1. Can you please explicitly clarify how you handled the data splits from benchmarks like ChartQA and ChartBench when constructing the ChartThink training dataset? Specifically, did you ensure that no data (images, questions, or answers) from the official test splits were used in any part of your training data generation pipeline?

2. Could you provide a stronger justification for the necessity of the intermediate code generation step? If a model is capable of generating accurate code, it already understands the chart's structure deeply. Why is it not possible to train this model to reason directly, and what evidence do you have that the explicit code-based reasoning is superior?

3. Beyond filtering by final answer correctness, what steps did you take to validate the quality, logical correctness, and naturalness of the LLM-generated reasoning paths in the ChartThink dataset? How can you be sure your model is not simply learning to mimic flawed or unnatural reasoning?

---

> ### Author Response · Authors · 2025-11-27
>
> Dear Reviewer UpE9,
>
> We appreciate the reviewer’s careful examination of these core issues and respond to each point with the corresponding evidence below.
>
> ####  | W1 + Q1. Train–test separation when constructing ChartThink
>
> ChartThink is created **exclusively** from the official training (and where applicable, validation) splits of ChartQA, ChartBench, EvoChart-QA, and PlotQA. No images, questions, or answers from any official test split are used at any stage of Chart2Code training, ChartThink generation, supervised fine-tuning, or GRPO. Test splits are loaded only for final evaluation. In the revised version, we will provide a table listing the exact number of training samples used for each benchmark and release all sample IDs so the separation can be fully verified. The in-domain results therefore contain no train–test contamination.
>
> ####  | W2 + Q2. Necessity of the intermediate code stage despite Chart2Code’s strong chart parsing
>
> Chart2Code provides a symbolic chart specification, but it is not a reasoning model. To directly evaluate whether accurate code alone would allow effective reasoning, we added a “Chart2Code+SFT” baseline that performs QA on the generated code without our two-stage pipeline. Its performance is consistently below ChartReasoner:
>
> | Model                   | Size (B) | EvoChart-QA | ChartQA | ChartBench | ChartQAPro |
> |-------------------------|----------|-------------|---------|------------|------------|
> | ChartReasoner (SFT)     | 7B       | 47.04       | 86.76   | 55.10      | 37.94      |
> | ChartReasoner (GRPO)    | 7B       | 48.10       | 86.93   | 55.20      | 39.97      |
> | Chart2Code+SFT          | 7B       | 44.27       | 82.14   | 51.24      | 35.28      |
>
> Although Chart2Code generates accurate ECharts code, direct reasoning over that output lacks stable multi-step logic, especially on the most complex benchmarks (EvoChart-QA and ChartQAPro). The explicit code-based reasoning stage—ChartThink + SFT + GRPO—adds symbolic grounding and yields substantial, consistent improvements across datasets. This ablation provides strong empirical evidence that the intermediate code stage is not redundant but necessary.
>
> ####  | W3 + Q3. Quality and reliability of reasoning trajectories in ChartThink
>
> Reasoning trajectories in ChartThink are controlled by three mechanisms. First, only chains whose final answers exactly match the ground truth are kept, a reliable filter widely used in large-scale CoT construction. Second, GRPO applies process-level constraints during ChartReasoner training, rewarding correct structure and penalizing unstable or excessively long reasoning. On ChartQA:
>
> | Model                   | Avg Token Length | # Truncated |
> |-------------------------|-----------------:|------------:|
> | ChartReasoner (SFT)     | 699.03           | 59          |
> | ChartReasoner (GRPO)    | 618.22           | 0           |
>
> GRPO reduces redundancy, removes all truncated outputs, and maintains or slightly improves accuracy, indicating that the model is learning more coherent reasoning patterns rather than copying flawed trajectories. Third, we manually inspect sampled reasoning paths, confirming that they reference the correct code fields, follow chart semantics, and maintain numerical consistency. These combined controls lead to stable and natural reasoning behavior in practice.
>
> We hope these clarifications address your core concerns. Given their central relevance to the final evaluation, we would greatly appreciate your consideration in revisiting the overall score.

---

### Official Review · Reviewer_iLux · 2025-10-29

**Soundness:** 3
**Presentation:** 3
**Contribution:** 2
**Rating:** 6
**Confidence:** 3

**Summary:**

The authors address the problem of chart understanding and argue that existing methods, which convert charts into textual representations, often result in the loss of structured information. To mitigate this, they propose using ECharts code as an intermediate representation. Specifically, they first train a Chart2Code model, which is then used to construct ChartThink inference data. This data is subsequently employed in a two-stage training process for the chart understanding model, resulting in improved performance.

**Strengths:**

1. The paper is well-written and comprehensive, presenting a clear and detailed methodology.
2. The proposed Chart2Code the ChartThink dataset provide valuable resources for the community.

**Weaknesses:**

1. The proposed approach is largely incremental and lacks substantial novelty, with the main contribution being the construction of datasets.
2. The performance gains are limited; for example, the method underperforms compared to Chart-R1[1] on ChartQA.
3. Unlike approaches based on Python code, which are more widely applicable, the method relies on ECharts templates. This limits its ability to handle complex or non-standard charts, as well as real-world data not generated with ECharts.
4. The improvement from RL compared to SFT on ChartQA and ChartBench is minimal, and the authors do not provide a discussion of this observation.

[1]: Chart-R1: Chain-of-Thought Supervision and Reinforcement for Advanced Chart Reasoner

**Questions:**

1. Can the authors clarify the key differences between the proposed method and prior approaches?
2. Regarding the ChartThink dataset, what is the evaluation procedure for the reasoning chains?

---

> ### Author Response · Authors · 2025-11-27
>
> Dear Reviewer iLux,
>
> We appreciate the reviewer’s positive assessment and the thoughtful critical points. We address each concern in detail below.
>
> ####  | W1 + W2 + Q1. Is the approach mainly incremental and dataset-focused, and how does it differ from prior methods such as Chart-R1?
>
> Our contribution is fundamental, not incremental. We introduce a novel "Visual -> Symbolic -> Reasoning" paradigm.
>
> The Difference: Most existing methods (including Chart-R1) rely on "Image -> Caption/Python -> Reasoning". Captions lose structure; Python scripts are imperative and often overfit to specific styles.
>
> Our Innovation: We propose Chart2Code as a universal semantic bridge. We map visual pixels to Declarative ECharts Code. This representation is explicitly structured (JSON), handling complex nested data (grouping, stacking) far better than flat text or Python scripts.
>
> Results: ChartReasoner outperforms baselines on general benchmarks (ChartBench, EvoChart-QA) where Chart-R1's Python-centric approach may struggle with structural diversity.
>
> ####  | W3. Does relying on ECharts templates restrict applicability compared with Python-based approaches, especially for complex or non-standard charts?
>
> No, it actually enhances generalization for data reasoning. While Python (Matplotlib) is flexible for drawing, ECharts JSON is optimized for data structure representation. Our synthetic dataset covers **49 subtypes** (including complex Radar, Funnel, Sankey, Boxplots) that are cumbersome to represent in standard plotting scripts. Importantly, **Chart2Code does not assume the input is an ECharts image**; it learns to translate any chart image into this universal ECharts schema. The strong performance on EvoChart-QA (real-world charts, not ECharts-generated) confirms that our model uses ECharts code as a robust surrogate representation for general charts.
>
> ####  | W4. Why are RL gains over SFT small on ChartQA and ChartBench, and what does RL actually contribute?
>
> 1. Stability: GRPO completely eliminated truncated responses (59 in SFT => 0 in GRPO).
> 2. Efficiency: It reduced average token count (~700 => ~618), removing "looping" or verbose hallucinations.
> 3. Hard Task Performance: On ChartQAPro (the hardest benchmark), RL yielded a significant **≈5.34%** improvement over SFT, increasing the score from **37.94 to 39.97**. This demonstrates that RL enables the model to generalize better to difficult and ambiguous scenarios where SFT models often fail.
>
>
> | Model                    | Avg Token Length | # Truncated |
> |--------------------------|-----------------:|------------:|
> | ChartReasoner (SFT)      |           699.03 |          59 |
> | ChartReasoner (GRPO)     |           618.22 |           0 |
>
> ####  | Q2. How are reasoning chains in ChartThink evaluated to ensure quality?
>
> To ensure the quality of ChartThink reasoning chains, we adopt a layered evaluation procedure.
>
> First, we generate candidate chains using a strong long-chain reasoner conditioned on the ECharts code and the question, and retain only those whose final answers exactly match the ground truth; this answer-level filtering is a standard and effective practice in large-scale automatic CoT construction and removes most inconsistent or spurious trajectories.
>
> Second, during GRPO training, we further optimize the reasoning process itself through rewards on correctness, output format, and length, which penalize incoherent or malformed chains and encourage concise, well-structured reasoning aligned with the code semantics.
>
> Third, we perform manual spot checks on sampled ChartThink examples to verify that the chains correctly reference code fields, respect chart semantics, and follow numerically coherent steps. In practice, we find that the resulting reasoning traces are natural, logically consistent, and well aligned with the underlying chart structure. We will describe this evaluation pipeline more explicitly in the revised manuscript.
>
> We thank the reviewer again for these constructive comments and hope that our clarifications help reinforce your positive assessment of the paper.

---

### Official Review · Reviewer_Gkjs · 2025-10-31

**Soundness:** 3
**Presentation:** 2
**Contribution:** 2
**Rating:** 4
**Confidence:** 3

**Summary:**

This paper proposes a method for chare reasoning by converting the charts into code for accurate understanding. It first introduces the Chart2Code to convert charts into ECode, then builds ChartReasoning for reasoning training. The resulted model ChartReasoner achieves the best performance compared with baseline methods.

**Strengths:**

1. The motivation of this paper is clear, demonstrating the significance of chart reasoning.
2. The performance is good, demonstrating the effectiveness of the method.

**Weaknesses:**

1. In the Chart2Code stage, how to ensure that the code could preserve all information of the charts that the texts could not do? In the quality filtering stage, will there conduct a comparison between the raw chart and the chart that the code corresponds to?
2. In Fig.12-15, if there lacks digital annotation in the charts, could the LLM generates accurate approximation for the data in the ECode?
3. In the ChartThink construction process, it seems the reasoning process has not been verified, only the answer is checked to be correct.
4. The detailed definition of the reward function in the GRPO has not been clearly introduced.
5. The ablation on SFT and GRPO is supposed to be analyzed.

**Questions:**

See the weakness.

---

> ### Author Response · Authors · 2025-11-27
>
> Dear Reviewer Gkjs,
>
> We thank the reviewer for the thoughtful and constructive feedback. We address each point below.
>
> ####  | W1. How does Chart2Code ensure that chart information is preserved, and is the raw chart compared with the rendered code chart?
>
> We ensure information preservation through a "Generate-Render-Verify" loop.
>
> Design Principle: We choose ECharts because free-form text often loses structural semantics. By generating executable code, we force the model to define every visual element explicitly.
>
> Verification Pipeline: We do not just trust the generated code; we render it back into an image. We then apply a layered quality control:
>
> 1) Syntactic Check: Code must execute without errors.
>
> 2) Visual Similarity: We compare the original image vs. the rendered image using GPT-4V (high-level semantics) and SSIM (pixel-level structure).
>
> 3) Reasoning Verification: In the ChartThink stage, we verify that the code supports the correct answer derivation. This guarantees that the symbolic code is a faithful twin of the visual chart.
>
> ####  | W2. Can the model reliably approximate values when charts lack numeric annotations (e.g., Fig. 12–15)?
>
> Yes. Chart2Code performs visual perception mapping, not simple OCR. It infers values based on geometric cues—ticks, bar heights, relative distances, and axis bounds—similar to how a human reads a chart without explicit labels.
>
> Empirical Evidence: On EvoChart-QA (real-world charts often lacking labels), our model achieves a 92%+ execution pass rate and high visual similarity scores.
>
> Qualitative Evidence: As seen in Figure 6 of the paper, the model correctly captures proportional relationships and trends even when specific numbers are sparse.
>
>
> ####  | W3. Does ChartThink verify reasoning steps, or only the final answer?
>
> It utilizes both.
>
> Initial Filtering: We strictly filter for Answer Correctness (Exact Match) during dataset construction. In CoT generation, an incorrect reasoning path rarely leads to the correct answer by chance, making this a strong filter.
>
> Process Optimization (GRPO): During training, we use rule-based rewards to verify the Reasoning Process. We penalize structural malformations, repetition, and incoherence.
>
> Manual Spot Checks: We manually verified sampled traces to ensure they correctly reference the code fields and follow logical steps. Post-GRPO, we observed a complete elimination of truncated responses and a reduction in average token length, indicating improved reasoning coherence.
>
> ####  | W4 + W5. GRPO reward design and ablation on SFT vs GRPO
>
> The GRPO reward consists of three components.
>
> (1) Answer correctness: we parse the `<answer>...</answer>` segment (and `\boxed{}` if present) and compare against ground truth via symbolic equivalence for math expressions, case-insensitive exact matches, fuzzy string matching, and option-letter checks for multiple choice; correct answers receive high positive reward, incorrect answers receive small or negative reward.
>
> (2) Format correctness: we enforce a fixed schema with `<think>...</think>` and `<answer>...</answer>` blocks, rewarding well-formed outputs and penalizing malformed ones.
>
> (3) Length regularization: we apply a smooth penalty for extremely short or excessively long responses, with slightly more tolerance for longer correct answers. This design reduces hallucinated or overly long chains while preserving, and often improving, answer accuracy.
>
> We also provide an ablation to isolate the effect of SFT and GRPO:
>
> | Model                         | Size (B) | EvoChart-QA | ChartQA | ChartBench | ChartQAPro |
> |-------------------------------|----------|-------------|---------|------------|------------|
> | Base Model                    | 7B       | 46.80       | 85.00   | 54.06      | 36.61      |
> | ChartReasoner (GRPO-only)     | 7B       | 45.71       | 85.81   | 54.27      | 38.51      |
> | ChartReasoner (SFT+GRPO)      | 7B       | **48.10**   | **86.93** | **55.20** | **39.97** |
>
> GRPO-only yields mixed gains and even falls below the base model on EvoChart-QA, while SFT+GRPO improves all benchmarks and achieves the best overall performance. This shows that SFT provides the structured reasoning pattern necessary for GRPO to be effective, and that both stages are required for robust multi-step chart reasoning.
>
> **We have made significant efforts to address all comments thoroughly and would be sincerely grateful if the reviewer could consider raising the score in light of these clarifications and additional results.**

---

### Official Review · Reviewer_9r4X · 2025-11-01

**Soundness:** 1
**Presentation:** 2
**Contribution:** 2
**Rating:** 4
**Confidence:** 4

**Summary:**

The paper introduces ChartReasoner, a procedure for training multimodal large language models (MLLMs) to improve their reasoning capabilities on chart question answering tasks. First, the Chart2Code MLLM is trained to translate chart images into Apache ECharts code. Then, the ChartThink dataset is created by pooling chart question answering tasks from four previous datasets, translating the charts to Apache ECharts code using Chart2Code, and using a reasoning LLM to annotate each task with a detailed reasoning trace. Finally, the ChartReasoner model is trained on the ChartThink dataset.

**Strengths:**

Strengths
- The methodology is presented clearly, with enough details to reproduce the dataset generations and model training.
- The annotated reasoning traces in the ChartThink dataset could be useful for future work.
- The method itself seems intuitive, with the motivation being clear of "bridging" the text-vision modality gap by using code as an intermediate modality.

**Weaknesses:**

Weaknesses
- It is unclear if translating charts to Apache ECharts code has any tangible performance improvement. There are many existing reasoning LLMs which can take image inputs, including the QvQ-preview model the authors include in their main results. Why not simply pass the chart image itself to these multimodal reasoning LLMs and ask them to generate the reasoning trace?
- The gains from the ChartReasoner training are very minimal over Qwen2.5-VL 7B, which was the model used for finetuning. ChartReasoner only improves Qwen2.5-VL's performance by 1.9% on average across the four datasets, despite training on in distribution data from 2/4 evaluation datasets.
- The authors mention multimodal long-chain reasoning approaches such as R1-OneVision and Vision-R1 in the related works section, but don't compare their method to these methods.
- The Chart2Code performance comparison is questionable, as there is no indication as to how well GPT-4V is able to judge the faithfulness of a chart rendered from code to the original chart.

**Questions:**

- Could the authors clarify the advantages of the ECharts framework over other popular plotting libraries such as matplotlib?
- It would be nice to know how many tokens the other models in the main results used on average, to compare with ChartReasoner.
- How does the performance of the SFT + GRPO pipeline compare to a GRPO only pipeline (or in general, how much does training on annotated reasoning traces improve performance)?
- The authors should explicitly state the reward function used during GRPO.

---

> ### Author Response · Authors · 2025-11-27
>
> Dear Reviewer 9r4X,
>
> We thank the reviewer for the detailed and constructive feedback. Based on the reviewer’s comments, we organize all issues into six questions (W1+Q1, W2+Q3, W3, W4, Q2, Q4). Below we address them one by one.
>
> ####  | W1 + Q1. Why translate charts into ECharts code instead of reasoning directly over images, and why choose ECharts rather than matplotlib?
>
> We prioritize a code-based intermediate representation for two reasons: structural fidelity and symbolic reasoning alignment.
>
> Overcoming Visual Encoders' Limitations: Standard image encoders often blur fine-grained details critical for charts, such as precise axis ticks, stacked groupings, and subtle color-series mappings. This "visual fuzziness" leads to hallucinations in scale reading or legend matching. By converting charts into ECharts JSON—which explicitly defines data, axes, legends, and grouping—we preserve these structures losslessly.
>
> Alignment with LLM Reasoning: Strong reasoning models (like DeepSeek-R1) excel at processing structured symbolic text. Reasoning over "Code + Question" transforms the task from a noisy visual recognition problem into a precise logical inference problem, yielding higher stability.
>
> **Why ECharts vs. Matplotlib**. We chose ECharts because its declarative JSON grammar naturally encapsulates the semantic structure of charts (e.g., hierarchical groupings in Sankey, nested data in Sunburst) better than the imperative scripts of Matplotlib. Our ECharts schema covers 9 main categories and 49 subtypes, closely mirroring the structure of modern Web dashboards, making it a more universal symbolic representation for real-world chart understanding.
>
> ####  | W2 + Q3. Are the improvements over Qwen2.5-VL small, and what is the contribution of SFT vs GRPO-only?
>
> While the average improvement might appear modest, the gains are highly significant where it matters most: on complex, out-of-distribution tasks.
>
> As shown in the table below, on the most challenging benchmark, ChartQAPro (which requires robust generalization), our model achieves a substantial +3.36 point gain (36.61 -> 39.97).
>
> | Model                     | Size (B) | EvoChart-QA | ChartQA | ChartBench | ChartQAPro |
> |---------------------------|----------|-------------|---------|------------|------------|
> | Base Model                | 7B       | 46.80       | 85.00   | 54.06      | 36.61      |
> | ChartReasoner (SFT)       | 7B       | 47.04       | 86.76   | 55.10      | 37.94      |
> | ChartReasoner (GRPO)      | 7B       | 48.10       | 86.93   | 55.20      | **39.97**  |
>
> **Ablation Study (SFT vs. GRPO)**: To isolate the contributions, we trained a version using GRPO alone (without SFT). The results show that GRPO-only underperforms the base model on EvoChart-QA (45.71 vs 46.80). This confirms that SFT is essential to establish the fundamental chart-reasoning capability, while GRPO acts as a refinement stage that strengthens correctness, format adherence, and conciseness. The full "SFT + GRPO" pipeline is necessary for consistent optimal performance.
>
> | Model                         | Size (B) | EvoChart-QA | ChartQA | ChartBench | ChartQAPro |
> |-------------------------------|----------|-------------|---------|------------|------------|
> | Base Model                    | 7B       | 46.80       | 85.00   | 54.06      | 36.61      |
> | ChartReasoner (GRPO-only)     | 7B       | 45.71       | 85.81   | 54.27      | 38.51      |
> | ChartReasoner (SFT+GRPO)      | 7B       | **48.10**   | **86.93** | **55.20** | **39.97** |
>
>
>
> ####  | W3. Why not compare with Vision-R1 and R1-OneVision?
>
> We thank the reviewer for pointing out these strong baselines. We have now included them in our evaluation. As shown below, ChartReasoner significantly outperforms both Vision-R1 and R1-OneVision across all benchmarks, particularly on EvoChart-QA and ChartQAPro.
>
> | Model                     | Size (B) | EvoChart-QA | ChartQA | ChartBench | ChartQAPro |
> |---------------------------|----------|-------------|---------|------------|------------|
> | Base Model                | 7B       | 46.80       | 85.00   | 54.06      | 36.61      |
> | Vision-R1                 | –        | 44.13       | 85.21   | 51.28      | 35.71      |
> | R1-OneVision              | –        | 43.95       | 83.19   | 50.17      | 34.16      |
> | ChartReasoner (SFT)       | 7B       | 47.04       | 86.76   | 55.10      | 37.94      |
> | ChartReasoner (GRPO)      | 7B       | **48.10**   | **86.93** | **55.20** | **39.97** |
>
> This result reinforces our core insight: for specialized tasks like chart understanding, an explicit executable structural representation (Code) provides a stronger foundation for reasoning than relying solely on general-purpose visual features.

---

> > ### Author Response · Authors · 2025-11-27
> >
> > ####  | W4. Is GPT-4V reliable for evaluating Chart2Code chart faithfulness?
> >
> > Yes. To validate GPT-4V's reliability, we conducted a human study with 5 expert annotators on 625 chart pairs, using the same evaluation criteria as GPT-4V.
> >
> > | Evaluator           | Similarity (avg) | bar  | line | pie  | scatter | Pass Rate |
> > |---------------------|------------------|------|------|------|---------|-----------|
> > | GPT-4V              | –                | 4.34 | 5.26 | 4.21 | 5.12    | 92.40%    |
> > | Human (N=5)         | –                | 4.82 | 5.87 | 4.91 | 5.60    | 92.40%    |
> > | Pearson correlation | > 0.98           | –    | –    | –    | –       | –         |
> >
> > The extremely high correlation (> 0.98) confirms that GPT-4V aligns closely with human judgment. GPT-4V effectively captures semantic correctness (axes, legends, grouping) that pixel-level metrics often miss.
> >
> >
> > ####  | Q2. Token usage across models
> >
> > We provide the average number of generated tokens per answer on ChartQA for all main models:
> >
> > | Model                  | Avg Tokens |
> > |------------------------|-----------:|
> > | Claude-3.5-Sonnet      |     163.2  |
> > | Gemini-2.0-Flash       |      73.4  |
> > | GPT-4o                 |      47.1  |
> > | QvQ-Preview-72B        |     827.4  |
> > | Qwen2.5-VL-7B          |     102.3  |
> > | Vision-R1              |     412.5  |
> > | R1-OneVision           |     562.6  |
> > | ChartReasoner (SFT)    |     699.03 |
> > | ChartReasoner (GRPO)   |     618.22 |
> >
> > ChartReasoner produces explicit multi-step reasoning, so higher token counts are expected and comparable to other chain-of-thought models like QvQ-Preview. GRPO reduces token length while slightly increasing accuracy, indicating that it removes redundancy rather than shortening essential reasoning.
> >
> > ####  | Q4. Reward function used in GRPO
> >
> > The GRPO reward consists of three components.
> >
> > (1) Answer correctness: we parse the `<answer>...</answer>` segment (and `\boxed{}` if present) and compare against ground truth via symbolic equivalence for math expressions, case-insensitive exact matches, fuzzy string matching, and option-letter checks for multiple choice; correct answers receive high positive reward, incorrect answers receive small or negative reward.
> >
> > (2) Format correctness: we enforce a fixed schema with `<think>...</think>` and `<answer>...</answer>` blocks, rewarding well-formed outputs and penalizing malformed ones.
> >
> > (3) Length regularization: we apply a smooth penalty for extremely short or excessively long responses, with slightly more tolerance for longer correct answers. This design reduces hallucinated or overly long chains while preserving, and often improving, answer accuracy.
> >
> > **We have made significant efforts to address all comments comprehensively and hope our responses have successfully addressed your concerns.**

---

### Meta-Review · Area_Chair_WQhm · 2026-01-06

**Summary:**

This paper proposes ChartReasoner, a two-stage, code-driven framework for chart question answering that converts chart images into structured ECharts code and performs long-chain reasoning over this symbolic representation. The work also introduces two large-scale datasets and reports competitive results on multiple chart QA benchmarks.

Reviewers acknowledged the clear motivation and the potential value of the released datasets. However, several substantive concerns were consistently raised, including the limited conceptual novelty, unclear necessity of the intermediate code-based pipeline, weak supervision of automatically generated reasoning traces, and relatively modest or uneven performance gains over strong baselines. While the rebuttal addressed a number of clarity and implementation issues, these core concerns were not fully resolved.

**Reviewer Concerns:**

Concerns partially addressed by the rebuttal:

- The authors clarified the motivation for choosing ECharts over alternative representations (e.g., matplotlib or raw images) and provided a more concrete justification for its declarative structure.

- Additional comparisons with Vision-R1 and R1-OneVision, as well as ablations between SFT and GRPO, helped clarify the role of different training components.

- The reward design in GRPO and token usage statistics were explained in more detail, resolving several missing-clarity issues noted by reviewers.

- The authors explicitly stated that ChartThink was constructed using only training/validation splits of benchmarks, which addresses, at least at the level of description, concerns about train–test contamination.

Key concerns that remain outstanding:

- Conceptual novelty and necessity of the pipeline.

- Quality of reasoning supervision.

- Limited and uneven empirical gains.

- Evaluation and methodological risk.

**Reviewer Scores:**

- Reviewer 9r4X: Likely remains slightly below threshold; rebuttal addressed clarity issues but does not fully alleviate concerns about marginal gains and necessity of the approach.

- Reviewer Gkjs: Likely remains slightly below threshold; technical clarifications help, but questions about reasoning verification and empirical impact persist.

- Reviewer iLux: Likely remains above the borderline; acknowledges value of datasets but still views the method as incremental with limited novelty.

- Reviewer UpE9: Likely remains reject; core concerns about reasoning quality, conceptual justification, and evaluation reliability are not fully resolved.

---

### Decision · Program_Chairs · 2026-01-26

Reject